# The Targeted Degradation of BRAF V600E Reveals the Mechanisms of Resistance to BRAF-Targeted Treatments in Colorectal Cancer Cells

**DOI:** 10.3390/cancers15245805

**Published:** 2023-12-12

**Authors:** Abygail G. Chapdelaine, Geng Chia Ku, Gongqin Sun, Marina K. Ayrapetov

**Affiliations:** Department of Cell and Molecular Biology, University of Rhode Island, Kingston, RI 02881, USA; agchapdelaine@uri.edu (A.G.C.); gengchia_ku@uri.edu (G.C.K.)

**Keywords:** proteolysis targeting chimera, PROTAC, *BRAF* V600E mutation, colorectal cancer, triple-negative breast cancer, intrinsic resistance, mitogen-activated protein kinase pathway

## Abstract

**Simple Summary:**

The *BRAF* V600E mutation is frequently found in cancer. It activates the MAPK pathway and promotes cancer cell proliferation, making BRAF an excellent target for anti-cancer therapy. While BRAF-targeted therapy is highly effective for melanoma, it is often ineffective against other cancers. This study uses a proteolysis targeting chimera (PROTAC) to probe the role of BRAF V600E in colorectal and triple-negative breast cancer cells. The study reveals a diverse set of biochemical and proliferative responses to BRAF V600E degradation: some cancer cells are killed by BRAF degradation, while others utilize additional oncogenic drivers, such as Src kinase phosphatidylinositol 3-kinase, to resist the effect of BRAF degradation. These responses provide a mechanistic explanation for the efficacy of BRAF-targeted therapy for some cancers and the intrinsic resistance in others.

**Abstract:**

The *BRAF* V600E mutation is frequently found in cancer. It activates the MAPK pathway and promotes cancer cell proliferation, making BRAF an excellent target for anti-cancer therapy. While BRAF-targeted therapy is highly effective for melanoma, it is often ineffective against other cancers harboring the *BRAF* mutation. In this study, we evaluate the effectiveness of a proteolysis targeting chimera (PROTAC), SJF-0628, in directing the degradation of mutated BRAF across a diverse panel of cancer cells and determine how these cells respond to the degradation. SJF-0628 treatment results in the degradation of BRAF V600E and a decrease in Mek activation in all cell lines tested, but the effects of the treatment on cell signaling and cell proliferation are cell-line-specific. First, BRAF degradation killed DU-4475 and Colo-205 cells via apoptosis but only partially inhibited the proliferation of other cancer cell lines. Second, SJF-0628 treatment resulted in co-degradation of MEK in Colo-205 cells but did not have the same effect in other cell lines. Finally, cell lines partially inhibited by BRAF degradation also contain other oncogenic drivers, making them multi-driver cancer cells. These results demonstrate the utility of a PROTAC to direct BRAF degradation and reveal that multi-driver oncogenesis renders some colorectal cancer cells resistant to BRAF-targeted treatment.

## 1. Introduction

The rapidly accelerated fibrosarcoma (RAF) family of serine/threonine kinases, ARAF, BRAF, and CRAF, are core components of a signal transduction pathway that regulates the proliferation and survival of mammalian cells [1]. They transduce signals downstream of RAS to the mitogen-activated protein kinase (MAPK) cascade. Mutations in the *BRAF* gene can cause normal cells to become cancerous, accounting for 6–8% of all human cancers [2,3]. Although many forms of *BRAF* mutations have been identified, most are due to a V600E substitution, which renders BRAF constitutively active [4]. This mutation leads to aberrant activation of the MAPK pathway, promoting oncogenesis. The *BRAF* V600E mutation accounts for approximately 10% of all colorectal cancer [5], 2% of non-small cell lung cancer [6], 15.7% of all breast cancers [7], 50% of cutaneous melanoma and thyroid cancer [2], 6% of brain cancers [8], and 100% of hairy cell leukemia [9].

Inhibition of mutant BRAF often causes cancer cell toxicity due to the cell’s acquired dependency [10,11] on mutant BRAF. Thus, BRAF V600E is an excellent target for developing pharmacological inhibitors and therapeutic interventions in the cancers [12]. Three BRAF kinase inhibitors, vemurafenib [13], dabrafenib [14], and encorafenib [15], have been approved as a treatment for melanoma patients carrying the *BRAF* V600E mutation. Although these BRAF inhibitors are effective in initial treatments [16], their clinical benefits are restrained by the rapid emergence of acquired resistance [17] and desensitization to continuous treatment [18]. Acquired resistance is rarely the result of secondary mutations in *BRAF* that confer resistance to the inhibitors. Instead, resistance is caused by the utilization of a plethora of alternative signaling mechanisms, such as activated epidermal growth factor receptor (EGFR) [19], CRAF [20], alternative splicing variants of BRAF mRNA [21], COT/Tpl2 (encoded by *MAP3K8*) [22], or additional alterations to the MAPK pathway [23]. Alternative strategies for suppressing oncogenic BRAF could alleviate the acquired resistance [24].

Although BRAF inhibitors have been broadly effective in treating melanoma patients who have the *BRAF* V600E mutation [25,26], a much smaller portion of *BRAF*-mutated patients with other cancer types, such as colorectal cancer (CRC), respond to the same treatment [27]. One mechanism for this observed intrinsic resistance in other cancer types is the activation of the MAPK pathway by EGFR [23,28,29]. The co-administration of BRAF/MEK and EGFR inhibitors has successfully treated this type of CRC [30,31]. However, confirmed objective response rates to this regimen are only 20%, and the clinical benefit is not durable, with a median progression-free survival of only 4.3 months. Multiple other molecular mechanisms may contribute to intrinsic resistance in CRC and other cancers. The identification of such mechanisms and finding of alternative approaches to BRAF inhibition are needed.

Recently, a new technology called Proteolysis Targeting Chimeras (PROTACs) has been developed to target and degrade specific disease-causing proteins [32]. PROTACs are heterobifunctional small molecules that utilize the intrinsic ubiquitin-proteasome system to degrade the targeted proteins [33,34,35]. They consist of a motif that binds to a protein of interest (POI), a flexible linker, and a ligand for an E3 ubiquitin ligase. In addition to inhibiting the catalytic activities of POIs, PROTACs direct the polyubiquitination of a POI, leading to its degradation. The BRAF inhibitor vemurafenib has been coupled to a ligand for the von Hippel Lindau E3 ubiquitin ligase by a piperazine linker to generate a PROTAC, SJF-0628, which directs the degradation of activated BRAF V600E in cells [3,36]. It has been tested in several homozygous melanoma cell lines for *BRAF* V600E [3,36]. Although a high degree of selectivity was achieved in these cell lines, the majority of *BRAF* V600E-bearing cancers are heterozygous, and SJF-0628 has not been extensively tested in these cancer types due to ineffective responses to BRAF-targeted treatment [37,38,39]. Considering that other cancer types harboring the *BRAF* V600E mutation may not respond to BRAF-targeted therapy, how such cancer cells would respond to SJF-0628-directed BRAF degradation remains unanswered.

In this study, we assess the effectiveness of the PROTAC, SJF-0628, in the broader cellular background of heterozygous *BRAF* V600E-driven CRC and triple-negative breast cancer (TNBC) cell lines [11,40]. The results reveal that SJF-0628 can direct the degradation of BRAF V600E in all cell lines tested. Although the degradation of BRAF causes complete inhibition of cell viability in some cell lines, only partial or minor inhibition is achieved in others. Further study reveals that the cells partially inhibited by BRAF degradation, including most of the CRC cell lines, also contain other oncogenic drivers. Thus, these results provide a mechanistic explanation for why BRAF inhibition is not sufficient for treating some CRCs and call for a strategy to develop combination targeted therapies that can simultaneously block multiple oncogenic drivers.

## 2. Materials and Methods

### 2.1. Cell Lines, Media, and Drugs

The human TNBC cell line, DU-4475, and the human CRC cell lines, Colo-205, LS-411N, HT-29, and RKO, were purchased from ATCC (Manassas, VA, USA). Cells were cultured in RPMI-1640, McCoy 5A, and EMEM media, respectively, as previously described [11]. The kinase inhibitors, dasatinib (D-3307), dabrafenib (D-5678), and trametinib (T-8123), were purchased from LC Laboratories (Woburn, MA, USA). MK-2206 (HY-10358) was purchased from MedChemExpress (Monmouth Junction, NJ, USA). SJF-0628 (Catalog # 7463) was purchased from Bio-Techne/TOCRIS (Minneapolis, MN, USA).

### 2.2. Cell Culture and Viability Assays

Cell culture and viability assays were performed as described previously [41]. Cell viability was determined in 96-well plates using Biolog Redox Dye MA (for suspension cells) or MTT dye assay (Thermo Fisher Scientific, Waltham, MS, USA) (for adherent cells) per the manufacturer’s instructions. In the MTT assay, the absorbances at 490 nm and 750 nm were taken as indicators of cell viability. All cell growth and drug inhibition experiments were performed at least twice in triplicate.

### 2.3. Curve Fitting by the Hill Equation

Dose–response data were fitted to a three-parameter Hill equation in GraphPad 10 Prism to determine the IC_50_ and *I*_MAX_ values. Mean  ±  SD and unpaired *t*-tests were performed in GraphPad Prism.

### 2.4. Time Course Experiments

To determine the time-dependent effects of drug treatments on cell viability, cells were plated at 20,000 cells per well and treated with drugs at indicated concentrations, as described in Section 2.2. The cell viability was determined at 0 h, 24 h, 48 h, and 72 h after the treatments were initiated.

### 2.5. Cell Treatments and Western Blots

To determine the effects of SJF-0628, a protein kinase inhibitor, or a combination of inhibitors on the signaling proteins, cells were seeded at 70% confluency and treated with drugs at indicated concentrations for 1 h (protein kinase inhibitors) or 48 h (SJF-0628) under standard cell culture conditions. Western blotting was performed as described previously [11]. Membranes were imaged using LI-COR Odyssey CxL Imager and analyzed using Image Studio software (LI-COR Biosciences, Lincoln, NE, USA). All antibodies were purchased from Cell Signaling Technology (Danvers, MA, USA) or Thermo Fisher Scientific (Waltham, MA, USA). A list of antibodies with vendors and catalog numbers and the original images of Western blots are presented in Appendix A. The density of each protein band in the Western blots was measured using ImageJ 1.53K software [42]. Full Western blot images can be found in Appendix A.

## 3. Results

### 3.1. Treatment with SJF-0628 Caused Specific Degradation of BRAF in All Cell Lines

The PROTAC SJF-0628 has been previously evaluated in melanoma cancer cells that are homozygous for *BRAF* V600E mutation [3,36]. To assess the effectiveness of SJF-0628 in a broader set of cancer cells representative of diverse biochemical and genetic backgrounds, *BRAF* genotype, and tumor types, we selected several *BRAF* V600E-containing CRC and TNBC cell lines (Table 1). According to the Catalogue Of Somatic Mutations In Cancer (COSMIC) [43], all cell lines in this study are heterozygous in the *BRAF* allele, with one copy containing a V600E mutation and the other being wild-type. These cell lines also contain various other oncogenic mutations. In addition to the V600E mutation, *BRAF* in HT-29 also contains a T119S mutation, but this mutation has not been biochemically characterized and it is unknown if it contributes to BRAF activation.

We first examined the effect of SJF-0628 on BRAF degradation in DU-4475, a TNBC cell line dependent on BRAF V600E for cell survival and proliferation [11]. The cells were treated with increased concentrations of SJF-0628 for 1 h, 24 h, or 48 h, and the total and V600E BRAF levels were determined by Western blots. The treatments caused concentration-dependent and time-dependent decreases in total and V600E BRAF (Figure 1A). The treatments at 1 μM for 24 and 48 h were more effective, causing a virtual complete loss of V600E BRAF and significant loss of total BRAF. The degradation appears more specific for the V600E variant, as there was a more preferential loss of the mutant variant than total BRAF.

Corresponding to the decrease in BRAF at 1 μM SJF-0628, pMEK and pErk levels also decreased even though the total MEK and Erk protein levels remained unaffected (Figure 1B). These results indicated that the degradation of BRAF resulted in the deactivation of the MAPK pathway. Both SJF-0628 and dabrafenib, a specific BRAF inhibitor, resulted in a dose-dependent inhibition of the cell viability in DU-4475 (Figure 1C). Even though dabrafenib is more potent than SJF-0628, with IC_50_s of 2.4 nM for dabrafenib versus 163 nM for SJF-0628, both inhibitors resulted in a near-complete inhibition of cell viability at 1 µM. SJF-0628 also showed a 3.7-fold increase in potency compared to vemurafenib (IC_50_ = 507 ± 16 nM [11]). The cytotoxicity of SJF-0628 is consistent with its ability to direct BRAF V600E degradation.

In the four CRC cell lines tested, treatment at 1 µM SJF-0628 for 48 h also caused significant decreases in BRAF levels (Figure 2). These results demonstrate that SJF-0628 effectively directs specific degradation of active BRAF, as previously reported [3,36]. However, the effects of SJF-0628 treatment on the MAPK pathway status were cell line-dependent. In Colo-205 cells, the degradation of BRAF also resulted in a corresponding decrease in total MEK and phosphorylated MEK levels. The decrease in MEK protein levels suggests that BRAF degradation also resulted in MEK degradation, consistent with PROTACs’ ability to co-degrade interacting partners via bystander ubiquitination [44]. The SJF-0628 treatment did not decrease MEK protein levels in other cell lines. It is unclear why MEK is co-degraded with BRAF only in Colo-205 cells. In LS-411N, HT-29, and RKO cells, the degradation of BRAF also resulted in a concomitant decrease in pMEK and pErk levels without affecting the total MEK and Erk protein levels.

### 3.2. Treatment with SJF-0628 and the BRAF Inhibitor Dabrafenib Caused Cell Line-Dependent Effects on Cell Viability

We then determined the effects of SJF-0628 treatment on the viability of the panel of cell lines and compared the impact to the inhibition by dabrafenib (Figure 3). The dose-response data were fitted to the Hill equation, and inhibitory parameters, including the IC_50_ and the maximal inhibition, were determined (summarized in Table 2). In comparing the responses of various cell lines to these treatments, two characteristics became apparent. First, the levels of maximal inhibition by dabrafenib and SJF-0628 on each cell line were in excellent agreement, suggesting that both drugs achieved their inhibition by blocking the same BRAF enzyme, albeit by different mechanisms. Second, the levels of maximal inhibition by either drug vary highly among the cell lines. Either drug virtually completely inhibited the viability of DU-4475 and Colo-205, while the viability of RKO and HT-29 was inhibited by only about 50% and 60%, respectively. While it is possible that in different cell lines, BRAF displays varying sensitivity to these treatments, the fact that the viability inhibition by both drugs reached a plateau near 1 µM in each cell line suggests that near-saturating inhibition was achieved.

Furthermore, the treatment of RKO with 1 μM SJF-0628 for 48 h caused BRAF to decrease to a nearly undetectable level, further indicating that the lack of complete viability inhibition was not due to insufficient degradation of BRAF. The most plausible explanation for the varied levels of inhibition is that the blockade of BRAF is only capable of causing a partial inhibition of cell viability in RKO, HT-29, and LS-411N, suggesting that other oncogenic drivers contribute to the cells’ survival and proliferation. This interpretation suggests that *BRAF* V600E-containing cell lines can be divided into two categories. The first category of cells, including DU-4475 and Colo-205, are killed by BRAF degradation or inhibition, because *BRAF* is the only oncogenic driver. These cancer cell lines can be considered mono-driver (*BRAF* V600E) cancer cells. The second category of cells, including RKO, HT-29, and LS-411N, are only partially inhibited by BRAF degradation or inhibition, because they likely have other oncogenic drivers besides *BRAF* V600E. These cells are likely multi-driver cancer cells. Cancers effectively treated by BRAF inhibition, such as most melanomas, probably belong to the first category, while other cancers refractory to BRAF-targeted treatment likely belong to the second category.

### 3.3. Treatment with SJF-0628 Causes Cell Apoptosis in DU-4475 and Colo-205 Cancer Cells

To further explore the possibility that other oncogenic drivers contribute to the varying levels of inhibition upon SJF-0628 and dabrafenib treatment, we first examined the responses of DU-4475 and Colo-205. DU-4475 was derived from a cutaneous metastatic nodule from a patient with advanced TNBC [45,46], and we have previously shown that BRAF V600E small molecule inhibitors, such as dabrafenib and vemurafenib, inhibit the cell viability of DU-4475 to completion, shut down the MAPK pathway, and induce apoptosis [11]. Based on these results, we used DU-4475 as a BRAF V600E-dependent cell line benchmark for evaluating the effects of SJF-0628. We compared how SJF-0628 and dabrafenib treatments affect cell viability over time (Figure 4A). The treatments of DU-4475 cells with 1 µM dabrafenib or 1 µM SJF-0628 caused a time-dependent loss of cell viability, reducing the cell viability to near background levels by 72 h. The decrease in relative viability below the initial seeding indicates that the treatment killed cells. The effect of dabrafenib was more immediate, as the cell viability started to decrease within 24 h, whereas the cell-killing effect of SJF-0628 was delayed and became evident after 48 h. This delay is likely due to the completion time for the degradation of BRAF, while dabrafenib can inhibit BRAF activity immediately. When DU-4475 cells were treated with SJF-0628 for 48 h, cleavage of poly (ADP-ribose) polymerase (PARP), which suppresses DNA repair, and caspases was observed (Figure 4B). These results confirmed that the targeted degradation of a single oncogenic driver, *BRAF* V600E, is sufficient to inhibit cell proliferation and induce apoptosis in DU-4475. This result is consistent with a previous study demonstrating DU-4475 as a mono-driver cancer cell line dependent on *BRAF* V600E [11].

Colo-205 is a colorectal cancer cell line that is also highly sensitive to BRAF inhibition and degradation (Figure 3A). As shown in Figure 4C, treatment of Colo-205 with 1 μM SJF-0628 resulted in a time-dependent decrease in cell viability after the initial delay, indicating that inhibition or degradation of BRAF V600E is lethal to Colo-205 cells. Western blots also detected the cleavage of PARP and caspases due to SJF-0628 or dabrafenib treatments (Figure 4D). Comparison of cell morphology in control cells and cells treated with SJF-0628 or dabrafenib indicates that the cells are indeed killed by the treatments (Figure 4E). These results demonstrate that *BRAF* V600E is the predominant driver in Colo-205, making it a mono-driver cancer cell line.

### 3.4. HT-29 and RKO Are Multi-Driver Cancer Cell Lines and Are Not Lethally Inhibited by BRAF Degradation or Inhibition

The partial inhibition of HT-29 and RKO viability by SJF-0628 and dabrafenib suggests that blocking BRAF function may not be sufficient to fully inhibit these cells’ viability. HT-29 has been shown to overexpress Src kinase [43], causing Src activation [40,47]. Thus, HT-29 contains at least two independent oncogenic drivers: *BRAF* and Src. We examined if Src is responsible for the resistance to BRAF inhibition or degradation. Dasatinib, a Src inhibitor, also partially inhibited HT-29 viability (Figure 5A,B), indicating that Src is a functional driver in HT-29 cells. The combination of dasatinib with SJF-0628 (Figure 5A) or dabrafenib (Figure 5B) was indeed much more effective than either drug alone, causing a much higher level of maximal inhibition and resulting in nearly complete inhibition at 1 µM. The combination index (CI) at 50% inhibition for SJF-0628 and dasatinib was 0.085, indicating strong synergy. However, the synergy at 50% inhibition was not strong for dasatinib and dabrafenib (CI = 1.9). The synergy for the dasatinib and dabrafenib combination became pronounced at inhibition levels above 60%, as either drug alone only reached 60% maximal inhibition, while the combination reached > 80% inhibition. The CI values for >60% inhibition were not calculable, because the individual drugs did not reach 60% inhibition.

The combination of dasatinib (1 µM) with either SJF-0628 (1 µM) or dabrafenib (1 µM) also caused time-dependent decreases in cell viability (Figure 5C), indicating that the drug combinations were killing the HT-29 cells. Western blotting confirmed that the combinations were much more effective in inducing the cleavage of PARP and caspases, suggesting that HT-29 cells were killed by these combinations via apoptosis (Figure 5D). In contrast, any of these drugs alone did not cause the cell viability to decrease significantly and did not induce the cleavage of these apoptosis-associated enzymes. These results support the conclusion that HT-29 cells depend on BRAF V600E and Src kinase for cell viability.

RKO is also only partially inhibited by dabrafenib or SJF-0628 alone, suggesting that some resistance mechanism is at play. This cell line contains an H1047R mutation in *PIK3CA*, the gene encoding phosphatidylinositol 3-kinase (PI3K) catalytic subunit. This mutation is one of the most frequent mutations in cancer [39,48] and has been shown to activate the PI3K lipid kinase activity and the PI3K pathway [49,50]. Activation of the PI3K pathway would result in resistance to drugs targeting BRAF and the MAPK pathway.

To explore the potential resistance mechanism in RKO, we determined how Akt and MEK phosphorylation responded to dabrafenib and MK-2206, a specific Akt inhibitor (Figure 6A), leading to the following observations. First, dabrafenib inhibition of BRAF also decreased MEK and Erk phosphorylation. Second, Akt is constitutively phosphorylated in this cell line, and the phosphorylation level is unaffected by dabrafenib treatment. Third, Akt phosphorylation is highly sensitive to inhibition by MK-2206, as either MK-2206 alone or in combination with dabrafenib caused a concentration-dependent inhibition of pAkt levels.

To our surprise, despite the potent inhibition of Akt phosphorylation by MK-2206, this drug did not inhibit the cell viability in RKO up to 1 µM (Figure 6B). The addition of MK-2206 did not improve the inhibition by either dabrafenib or SJF-0628 (Figure 6B,C). Time courses of RKO proliferation in the presence of various inhibitors and inhibitor combinations revealed that some of the inhibitors and inhibitor combinations partially inhibited cell proliferation, but cell viability did not decrease over time under any of these treatments (Figure 6D). These results indicate that none of these treatments were effective in activating apoptosis in RKO. Western blots did not detect any cleavage of caspases 3 and 9 under any of these treatments (Figure 6E). The treatment with the combination of dabrafenib and MK-2206 produced some cleavage of PARP. These results suggest that other proliferation-stimulating mechanisms are still functional even when both BRAF V600E and Akt are inhibited.

To determine if residual MAPK pathway is responsible for RKO resistance to BRAF inhibition and degradation as well as Akt inhibition, we tested the effect of trametinib on RKO viability (Figure 7). Trametinib is a potent inhibitor of both MEK activity and phosphorylation [51,52,53]. Trametinib was indeed a much more potent inhibitor (IC_50_ > 0.01 nM) of RKO viability than either SJF-0628 or dabrafenib (IC50 < 1 µM for both) (Figure 7A). Furthermore, treatment with trametinib caused a marked decrease in MEK and Erk phosphorylation in RKO (Figure 7B). However, trametinib alone only inhibited RKO cell proliferation and did not cause a decrease in RKO cell viability over time (Figure 6C), indicating that blocking the MAPK pathway by trametinib is still insufficient to kill RKO cells, presumably because the PI3K pathway is activated by the H1047R mutation in *PIK3CA* and/or other oncogenic drivers.

We then determined if the combination of trametinib and MK-2206 could induce cell death in RKO (Figure 7C). The results indicated that the combination could inhibit RKO cell proliferation and cause a slight decrease in cell viability over time. This result suggests that the drug combination likely caused some level of cell killing by blocking both the MAPK kinase pathway and the PI3K pathway. A comparison of the morphologies of RKO cells treated with various drugs indicated that the combination killed some cells (Figure 7D). These results demonstrate that RKO cells contain multiple mechanisms supporting cell survival and proliferation, making this cell line a multi-driver cancer.

## 4. Discussion

BRAF is frequently activated by the V600E mutation in many cancers [2]. Targeting BRAF with small molecule inhibitors, such as dabrafenib and vemurafenib, is a successful therapeutic approach for melanoma [41]. However, the therapeutic benefit is often limited by acquired resistance. Furthermore, targeting BRAF in other cancer types harboring the *BRAF* V600E mutation has not been as effective as in melanoma, suggesting that these cancers are intrinsically resistant to BRAF inhibition [27]. Alternative approaches to blocking BRAF can overcome acquired and intrinsic resistance. PROTACs provide a new strategy to eliminate the function of a target protein via degradation rather than solely inhibiting its activity. They direct the polyubiquitination and proteasomal degradation of the target proteins and offer an effective tool to specifically and post-translationally manipulate the cellular concentration and function of proteins of interest [32]. In this study, we took advantage of a recently developed PROTAC, SJF-0628, to probe the role of BRAF V600E in CRC and TNBC cell lines.

### 4.1. SJF-0628 Causes Specific Degradation of BRAF V600E in CRC and TNBC Cancer Cells

SJF-0628 is a PROTAC based on vemurafenib. Our results show that SJF-0628 effectively directs the degradation of BRAF V600E in all cell lines tested, including four CRC cell lines and one TNBC cell line that are heterozygous at the *BRAF* allele. Compared to BRAF inhibition by small molecule inhibitors such as dabrafenib, SJF-0628 consistently generated the same level of maximal inhibition of cell viability, even though dabrafenib is a more potent inhibitor based on IC_50_s. This suggests that this PROTAC affected cell viability by degrading BRAF without causing other non-specific effects. The results demonstrated the utility of SJF-0628 in specifically directing BRAF degradation. As reported previously, SJF-0628 treatment resulted in the preferential degradation of the mutant and activated form of BRAF [3,36].

The PROTAC-directed degradation takes longer to achieve (24–48 h) than the inhibition by small molecule inhibitors (within 1 h). Nevertheless, the ability to specifically degrade BRAF V600E provides a convenient tool for probing the role of BRAF V600E in driving the proliferation of several heterozygous *BRAF* V600E-harboring cancer cell lines.

### 4.2. DU-4475 and Colo-205 Are BRAF V600E-Dependent Mono-Driver Cancer Cell Lines

The effects of BRAF degradation on cell viability divide *BRAF* V600E-harboring cancer cells into two broad categories: those killed by BRAF V600E degradation and those whose viability is only partially inhibited. The former category includes the TNBC cell line DU-4475 and the CRC cell line Colo-205, which are killed via apoptosis due to BRAF degradation or inhibition by either SJF-0628 or dabrafenib. Thus, these cells appear to be “addicted” to a single oncogenic driver, *BRAF* V600E, according to the oncogene addiction hypothesis [54,55]. Cancers utilizing such a mono-driver mechanism are likely responsive to targeted therapy in a clinical setting. Most melanoma cancers and some CRCs probably belong to this category.

Interestingly, although a *BRAF* V600E mutation is rare in TNBC, several TNBC patients harboring this mutation have been reported [56,57,58]. All of these patients responded positively to the initial treatment with BRAF-targeted therapy, but some acquired resistance during continued treatment. Currently, all cancer types effectively treated by targeted therapy appear to be mono-driver cancers, such as chronic myeloid leukemia dependent on BCR-Abl [59], non-small cell lung cancer dependent on mutated EGFR [60], and gastrointestinal stromal tumors dependent on c-Kit [61].

### 4.3. Most CRC Cells Are Not Killed by BRAF Degradation or Inhibition Due to Their Multi-Driver Nature

Among the four CRC cell lines, degradation or inhibition of BRAF is lethal to only one, Colo-205. For the other three CRC cell lines, HT-29, LS-411N, and RKO, BRAF inhibition and degradation only partially affected proliferation and did not induce cell-killing. Their continued survival and proliferation when BRAF is inhibited or degraded suggests that they contain other activated oncogenic drivers, making them multi-driver cancer cells. Further investigation into two cell lines, HT-29 and RKO, confirmed this hypothesis and revealed the mechanisms that conferred resistance to BRAF-targeted treatments.

In HT-29, the resistance to BRAF degradation or inhibition is due to the oncogenic contribution of another driver, Src kinase [40,46]. Src kinase is overexpressed in HT-29 cells [42]. The observation that dasatinib partially inhibits HT-29 viability suggests that Src is activated independently of BRAF degradation or inhibition. The current study demonstrates that combining the Src inhibitor dasatinib with SJF-0628 or dabrafenib is highly effective in killing HT-29 cells via apoptosis, while the individual drugs failed to do so.

RKO cells are another multi-driver cancer cell model resistant to BRAF treatment. Our results demonstrated that SJF-0628, dabrafenib, and trametinib all blocked MEK and Erk phosphorylation in RKO and shut down the MAPK pathway. Trametinib is a much more potent inhibitor of RKO viability than dabrafenib or SJF-0628. Furthermore, the PI3K pathway in RKO is activated due to an oncogenic mutation, H1047R, in *PIK3CA*, the gene encoding the p110 catalytic subunit of PI3K. The combination of trametinib and MK-2206, which block the MAPK pathway and Akt in the PI3K pathway, respectively, completely inhibits RKO proliferation and achieves some level of cell killing.

These results demonstrate the biochemical diversity and complexity of CRC, as each cell line presents a unique response to the degradation and inhibition of BRAF V600E. These results reveal multiple mechanisms for CRC cancer cells to resist the inhibition or degradation of BRAF. It is likely necessary to understand these resistance mechanisms and consider them in developing successful targeted therapy against CRC.

## 5. Conclusions

A major challenge in cancer research and targeted therapy is that the presence of an oncogenic driver does not always correlate with the effectiveness of a treatment targeting that driver. The *BRAF* V600E mutation is an excellent example of this challenge. It is well established that this mutation activates BRAF and is oncogenic, and targeting BRAF V600E is an effective treatment strategy in melanoma. However, this strategy does not work in most other cancers containing the same mutation. About 15% of CRC patients have the *BRAF* V600E mutation, yet they are mainly resistant to BRAF-targeted treatment.

This study demonstrated that a proteolysis targeting chimera (PROTAC), SJF-0628, is effective in directing the degradation of BRAF V600E in CRC and TNBC cancer cells, providing a tool to probe the effects of BRAF degradation and inhibition on cell signaling and proliferation. The study revealed a diverse set of cell responses to these perturbations, reflecting the varying roles of BRAF in oncogenesis, from serving as the sole driver in some mono-driver cancer cells to playing a complementary role in multi-driver cancer cells. The study identified several mechanisms that complement BRAF V600E-dependent proliferation in CRC oncogenesis and confer resistance to BRAF-targeted treatments. Understanding and considering these resistance mechanisms when designing targeted therapies are likely necessary to successfully implement targeted therapy in multi-driver cancers, such as CRCs.

## Figures and Tables

**Figure 1 cancers-15-05805-f001:**
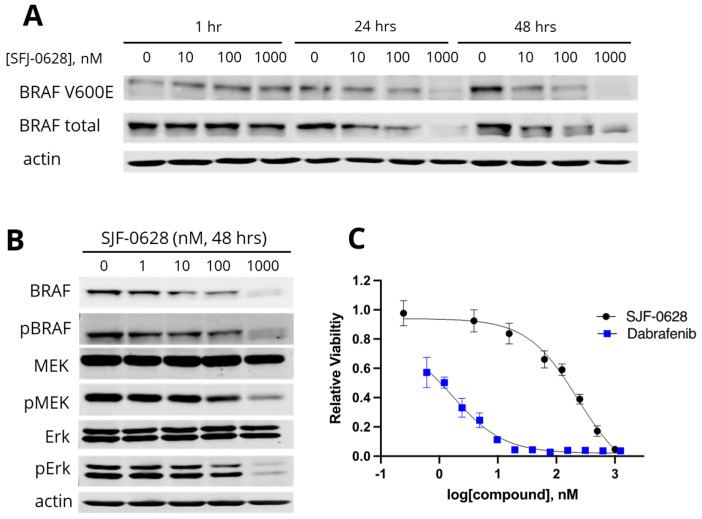
Effects of SJF-0628 on the BRAF level, MAPK pathway status, and cell proliferation of DU-4475 cells. (**A**). Time-dependent and concentration-dependent degradation of BRAF V600E and total BRAF upon SJF-0628 treatment. (**B**). Induced BRAF degradation and suppression of MEK and Erk phosphorylation. Cells were treated with indicated amounts of SJF-0628 for 48 h. (**C**). Cell proliferation assay in DU-4475 cells treated with increasing amounts of SJF-0628 and dabrafenib for 72 h (mean ± SD, n = 3 biologically independent samples); SJF-0628 = black, dabrafenib = blue.

**Figure 2 cancers-15-05805-f002:**
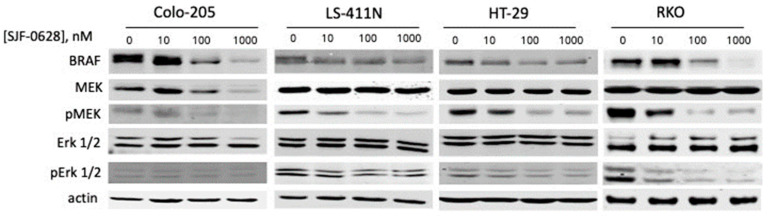
Effects of SJF-0628 treatments on the BRAF level and MAPK pathway status in four colorectal cancer cell lines. Four CRC cell lines were treated with increasing amounts of SJF-0628 for 48 h. Suppression of MEK and Erk phosphorylation was cell-line specific.

**Figure 3 cancers-15-05805-f003:**
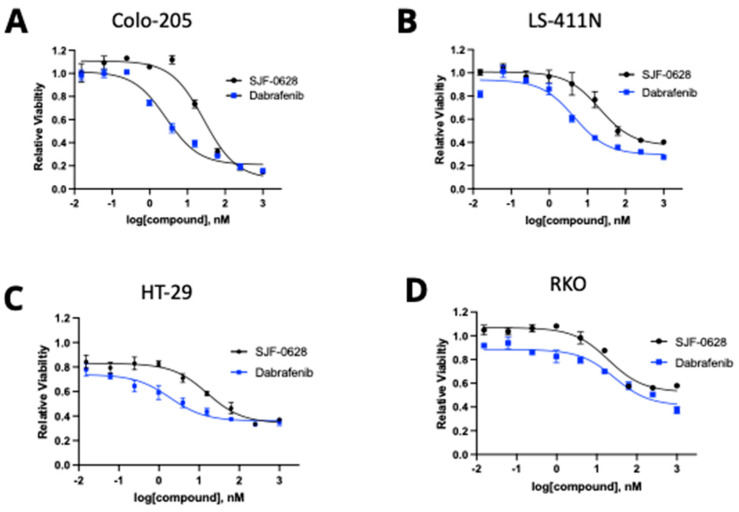
Effects of SJF-0628 and dabrafenib treatments on the proliferation of four colorectal cancer cell lines. Cells were treated with increasing amounts of SJF-0628 or dabrafenib for 72 h (mean ± SD, n = 3 biologically independent samples). (**A**) Colo-205. (**B**) LS-411N. (**C**) HT-29. (**D**) RKO. SJF-0628 = black, dabrafenib = blue.

**Figure 4 cancers-15-05805-f004:**
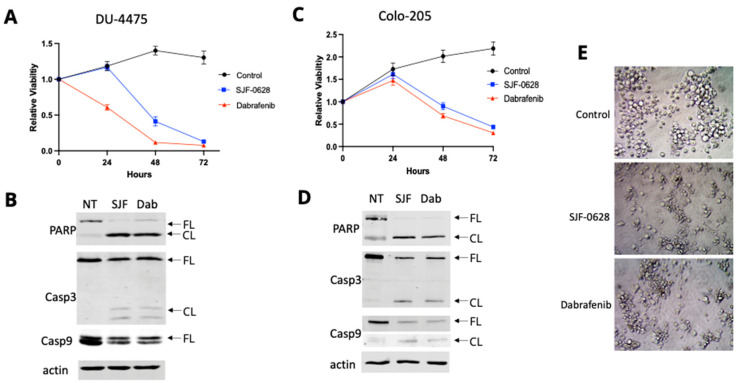
Effects of SJF-0628 and dabrafenib treatments on the proliferation and apoptosis of DU-4475 and Colo-205 cells. (**A**) Cellular proliferation of DU-4475 was monitored in the presence of 1 µM SJF-0628 or dabrafenib for 72 h (mean ± SD, n = 3 biologically independent samples). Control = black, SJF-0628 = blue, dabrafenib = red. (**B**) The presence of apoptotic cleavages of PARP, Casp-3, and Casp-9 was assessed via Western Blot in DU-4475 after 48 h treatment with 1 µM SJF-0628 or dabrafenib. (**C**) Cellular proliferation of Colo-205 was monitored in the presence of 1 µM SJF-0628 or dabrafenib for 72 h (mean ± SD, n = 3 biologically independent samples). Control = black, SJF-0628 = blue, dabrafenib = red. (**D**) The presence of apoptotic cleavages of PARP, Casp-3, and Casp-9 was assessed via Western blot in Colo-205 after 48 h treatment with 1 µM SJF-0628 or dabrafenib. (**E**) Morphological examination of Colo-205 after 48 h treatment with 1 µM SJF-0628 or dabrafenib.

**Figure 5 cancers-15-05805-f005:**
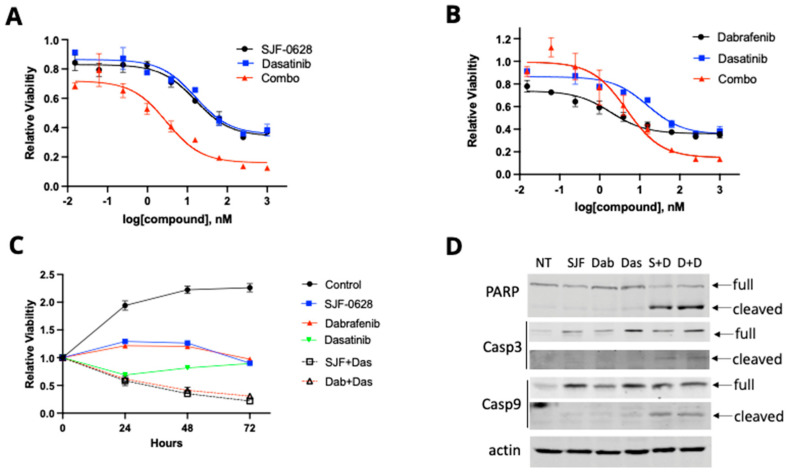
Effects of SJF-0628 and dabrafenib alone or in combination with dasatinib on HT-29 cells. (**A**) Cells were treated with increased amounts of SJF-0628, dasatinib, or a combination for 72 h (mean ± SD, n = 3 biologically independent samples). SJF-0628 = black, dasatinib = blue, combination = red. (**B**) Cells were treated with increased dabrafenib, dasatinib, or a combination for 72 h (mean ± SD, n = 3 biologically independent samples). Dabrafenib = black, dasatinib = blue, combination = red. (**C**) Cellular proliferation was monitored in the presence of 1 µM SJF-0628, dabrafenib, dasatinib, and their combination for 72 h (mean ± SD, n = 3 biologically independent samples). Control = black, SJF-0628 = blue, dabrafenib = red, dasatinib = green, SJF/dasatinib combo = black dash, dabrafenib/dasatinib combo = red dash. (**D**) The presence of apoptotic cleavages of PARP, Casp-3, and Casp-9 was assessed via Western blot after 48 h treatment with 1 µM SJF-0628, dabrafenib, dasatinib, and their combinations.

**Figure 6 cancers-15-05805-f006:**
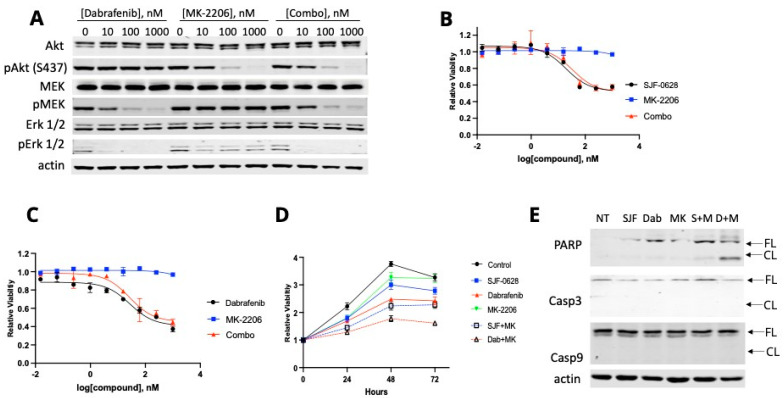
Effects of SJF-0628 alone or in combination with other inhibitors on RKO cells. (**A**) MEK, Erk, and Akt phosphorylation were assessed via Western blot after 1 h of treatment with increased amounts of dabrafenib, MK-2206, and their combination. (**B**) Cells were treated with increased amounts of SJF-0628, MK-2206, or a combination for 72 h (mean ± SD, n = 3 biologically independent samples). SJF-0628 = black, MK-2206 = blue, combination = red. (**C**) Cells were treated with increased dabrafenib, MK-2206, or a combination for 72 h (mean ± SD, n = 3 biologically independent samples. Dabrafenib = black, MK-2206 = blue, combination = red. (**D**) Cellular proliferation was monitored in the presence of 1 µM SJF-0628, dabrafenib, MK-2206, and their combination for 72 h (mean ± SD, n = 3 biologically independent samples). Control = black, SJF-0628 = blue, dabrafenib = red, MK-2206 = green, SJF/MK combo = black dash, dabrafenib/MK combo = red dash. (**E**) The presence of apoptotic cleavages of PARP, Casp-3, and Casp-9 was assessed via Western blot after 48 h treatment with 1 µM SJF-0628, dabrafenib, MK-2206, and their combinations.

**Figure 7 cancers-15-05805-f007:**
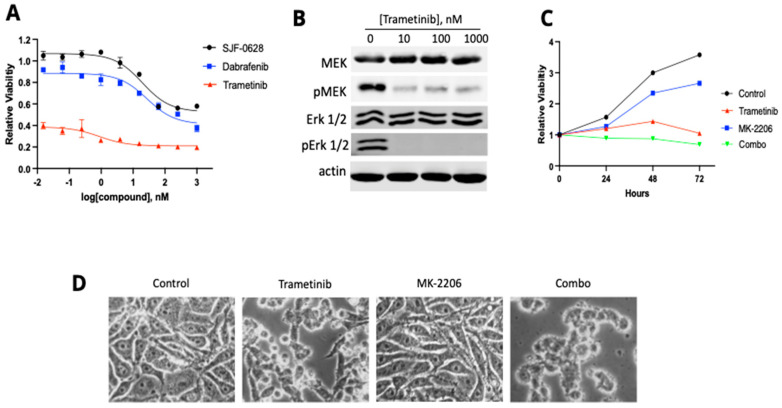
Effect of MEK inhibitor trametinib on RKO cell viability and the phosphorylation of MEK and Erk. (**A**) Cells were treated with increased amounts of SJF-0628, dabrafenib, or trametinib for 72 h (mean ± SD, n = 3 biologically independent samples. SJF-0628 = black, dabrafenib = blue, trametinib = red. (**B**) MEK, Erk, and Akt phosphorylation were assessed via Western blot after 1 h-treatment with increased amounts of trametinib. (**C**) Cellular proliferation was monitored in the presence of 100 nM trametinib or 1 µM MK-2206 and their combination for 72 h (mean ± SD, n = 3 biologically independent samples). Control = black, MK-2206 = blue, trametinib = red, SJF/MK combo = green. (**D**) Morphological examination of RKO after 72 h treatment with 100 nM trametinib, 1 µM MK-2206, and their combination.

**Table 1 cancers-15-05805-t001:** Cell lines ^1^ used in this study.

Cell Line	Tumor Type	*BRAF* V600E Genotype	Other Drivers	Cancer Mutations	Total Mutations
DU-4475	TNBC	Heterozygous	None	50	387
Colo-205	CRC	Heterozygous	None	42	312
LS-411N	CRC	Heterozygous	Unknown	567	7141
HT-29	CRC	Heterozygous	Src	24	676
RKO	CRC	Heterozygous	PI3K, others	155	4762

^1^ All information on these cell lines is taken from the Catalogue Of Somatic Mutations In Cancer (COSMIC) [43].

**Table 2 cancers-15-05805-t002:** Inhibition ^1^ parameters of SJF-0628 and dabrafenib in all tested cell lines.

Cell Line	IC_50_ (nM)		I_max_ (%)	
	SJF-0628	Dabrafenib	SJF-0628	Dabrafenib
DU-4475	163 ± 8.2	2.4 ± 0.5	91.5 ± 3.4	98.0 ± 0.25 *
Colo-205	37.6 ± 6.0	7.9 ± 4.1	85.2 ± 0.3	86.3 ± 2.9
LS-411N	96.3 ± 15.2	4.4 ± 2.5	65.2 ± 10.0	63.7 ± 8.0
HT-29	53.6 ± 5.4	3.82 ± 2.3	63.0 ± 1.2	65.6 ± 2.9
RKO	<1 µM	<1 µM	42.0 ± 0.2	62.8 ± 0.27

^1^ The inhibitory parameters were calculated by fitting nine concentration dose–response data to the Hill equation described in Material and Methods. Mean ± SD was calculated from two sets of independent assays done in triplicates. * Previously reported values [11].

## Data Availability

Data supporting the reported results are provided in Appendix A or available upon request.

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
