# Peer review of "The Targeted Degradation of BRAF V600E Reveals the Mechanisms of Resistance to BRAF-Targeted Treatments in Colorectal Cancer Cells"

_cancers, 2023, doi:10.3390/cancers15245805_

Round 1

Reviewer 1 Report

Comments and Suggestions for Authors

By using the small molecule SJF-0628 targeting BRAF, the authors investigated the degradation profile of BRAF in various cancer cells and tested cell viability and apoptosis, they found inhibition of cell viability differed even though SJF-0628 can cause degradation of BRAF in all tested cells, giving an explanation for the sometimes ineffectiveness of SJF-0628-based treatments. Results here can basically support their conclusions.

It has been convinced that SJF-0628 can degrade mutant BRAF without affecting WT BRAF, but this degradation is not only limited to V600E but also applied to other mutations, so it would be more appropriate if the authors could include this point when describing their results. Especially considering that they are using various kinds of cancer cells and most of these cells are heterozygous for the genotype of BRAF, which means a decrease in the total BRAF may not only come from the decrease of V600E.

Figure 4B: actin is missing

Author Response

Comment 1: It has been convinced that SJF-0628 can degrade mutant BRAF without affecting WT BRAF, but this degradation is not only limited to V600E but also applied to other mutations, so it would be more appropriate if the authors could include this point when describing their results. Especially considering that they are using various kinds of cancer cells and most of these cells are heterozygous for the genotype of BRAF, which means a decrease in the total BRAF may not only come from the decrease of V600E.

Response: We appreciate this comment. In response we added the following sentence (lines 152-154): In addition to the V600E mutation, BRAF in HT-29 also contains a T119S mutation, but this mutation has not been biochemically characterized and it is unknown if it contributes to BRAF activation.

Comment 2: Figure 4B: actin is missing.

Response: The actin bands were accidentally cut off in processing the figure. Now they are there.

We appreciate all the suggestions and comments from the reviewers and revised the manuscript accordingly. We hope that these revisions meet the approval of the reviewers and editors.

Reviewer 2 Report

Comments and Suggestions for Authors

A study by Chapdelaine et al reveals a diverse set of biochemical and proliferative responses to BRAF V600E degradation: some cancer cells are killed by BRAF degradation, while others utilize diverse mechanisms to resist the effect of BRAF degradation, such as continued activation of MEK in the absence of BRAF V600E, overexpression and activation of Src kinase, or activated phosphatidylinositol 3-kinase.

Overall, the manuscript shows technically valid results, and presents subtantial novelty in the field of BRAF V600 inhibitors. The results are presented clearly, and in a logical fashion. The conclusions are supported by the results. The manuscript is well written, and discussion of the results is sufficient. I have only two minot/technical issues.

1. Gene names should be written in italics.

2. Fig. 4B lacks actin image.

Author Response

Comment 1: Gene names should be written in italics.

Response: We revised it accordingly.

Comment 2: Fig. 4B lacks actin image.

Response: The actin bands were accidentally cut off in processing the figure. Now they are there.

We appreciate all the suggestions and comments from the reviewers and revised the manuscript accordingly. We hope that these revisions meet the approval of the reviewers and editors.

Reviewer 3 Report

Comments and Suggestions for Authors

The authors evaluated the efficacy of a proteolysis targeting chimera (PROTAC), SJF-0628 in degrading BRAF protein in different cancer cells. They found SJF-0628 can induce cell death, or inhibit cell proliferation in different cancer cell lines they examined. This work is potentially of interest to readers in cancer researchers. However, the authors may address following concerns to improve this manuscripts.

Major concerns:

1. In materials and methods part, there are missing information about the drugs and antibodies. Please indicate the catalogue number, dilution of the drugs and antibodies used in this study.

2. line 131, 5% DMSO may cause cytotoxicity when the cell viability assay is performed.

3 figure 3, please add the graph title (Cell line) in each panel which helps reader understanding.

4. Figure 4B actin result is missing.

5. Figure 5, add the combination index (synergistic effect) of the combo treatments.

Minor concerns

 line 76, delete the () (EGFR)

Comments on the Quality of English Language

Need to improve.

Author Response

Major concern 1. In materials and methods part, there are missing information about the drugs and antibodies. Please indicate the catalogue number, dilution of the drugs and antibodies used in this study.

Response: This is an excellent suggestion. We added catalog and vendor information for all drugs (lines 112-115) and put vendor and catalog information for all antibodies in the Supplementary Materials. The concentrations of drugs are given in the results. The antibodies were used in Western blots according to the manufacturer’s instructions.

Major concern 2. line 131, 5% DMSO may cause cytotoxicity when the cell viability assay is performed.

Response: The final concentration of DMSO in the assay is 1%. The 5% number given was the concentration of DMSO in the 30 ml solution added into assay. The total volume in each well was 150 ml, so the final DMSO concentration was 1%, which did not cause cytotoxicity. In revising the detailed description of the method was deleted, but we referred to previous publications for details (line 117).   

Major concern 3. figure 3, please add the graph title (Cell line) in each panel which helps reader understanding.

Response: Added accordingly.

Major concern 4. Figure 4B actin result is missing.

Response: The actin bands were accidentally cut off in processing the figure. Now they are there.

Major concern 5. Figure 5, add the combination index (synergistic effect) of the combo treatments.

Response: This is an excellent suggestion, and we added the following sentences in response (lines 293-299): “The combination index (CI) at 50% inhibition for SJF-0628 and dasatinib was 0.085, indicating strong synergy. However, the synergy at 50% inhibition was not strong for dasatinib and dabrafenib (CI=1.9). The synergy for the dasatinib+dabrafenib combination became pronounced at inhibition levels above 60%, as either drug alone only reached 60% maximal inhibition, while the combination reached >80% inhibition. The CI values for >60% inhibition were not calculable because the individual drugs did not reach 60% inhibition.” 

Minor concern. line 76, delete the () (EGFR).

Response: Deleted.

We appreciate all the suggestions and comments from the reviewers and revised the manuscript accordingly. We hope that these revisions meet the approval of the reviewers and editors.

Round 2

Reviewer 3 Report

Comments and Suggestions for Authors

The authors have adequately addressed my concerns.